# FEM Simulation of AlSi10Mg Artifact for Additive Manufacturing Process Calibration with Industrial-Computed Tomography Validation

**DOI:** 10.3390/ma16134754

**Published:** 2023-06-30

**Authors:** Cesare Patuelli, Enrico Cestino, Giacomo Frulla, Federico Valente, Guido Servetti, Fabio Esposito, Luca Barbero

**Affiliations:** 1Department of Mechanical and Aerospace Engineering (DIMEAS), Politecnico di Torino, Corso Duca degli Abruzzi 24, 10129 Torino, Italy; enrico.cestino@polito.it (E.C.); giacomo.frulla@polito.it (G.F.); 2ITACAe S.r.l, Via Calosso 3, 14100 Asti, Italy; f.valente@itacae.com (F.V.); g.servetti@itacae.com (G.S.); 3TEC Eurolab S.r.l, Viale Europa, 40, 41011 Campogalliano, Italy; esposito@tec-eurolab.com; 4SPEM S.r.l, Via Torino, 307, 10032 Brandizzo, Italy; barberol@ellenaspa.com

**Keywords:** selective laser melting, finite elements, industrial computed tomography, calibrating artifact

## Abstract

Dimensional accuracy of selective laser melting (SLM) parts is one of manufacturers’ major concerns. The additive manufacturing (AM) process is characterized by high-temperature gradients, consolidation, and thermal expansion, which induce residual stress on the part. These stresses are released by separating the part from the baseplate, leading to plastic deformation. Thermo-mechanical finite elements (FE) simulation can be adopted to determine the effect of process parameters on final geometrical accuracy and minimize non-compliant parts. In this research, a geometry for process parameter calibration is presented. The part has been manufactured and then analyzed with industrial computed tomography (iCT). An FE process simulation has been performed considering material removal during base plate separation, and the computed distortions have been compared with the results of the iCT, revealing good accordance between the final product and its digital twin.

## 1. Introduction

Lightweight constructions can reduce fuel consumption and thus ${\mathrm{CO}}_{2}$ emissions, which is a key point for the aerospace and automotive industry. The importance of the role of topology optimization is increasing as it allows us to optimize the geometry of the parts and achieve a weight reduction. However, the complexity of the parts obtained with optimization or generative design is often not suitable or cost intensive for traditional manufacturing technologies like casting, extrusion, or machining. Lattice structures and volumes with cavities are examples of geometries that can be manufactured only with additive manufacturing (AM) technologies [1] such as selective laser melting (SLM). AM allows the manufacturing of these geometries with high precision by significantly enlarging the design space. On the other hand, SLM can be considered as a series of micro-welding processes and carries the same problems in terms of residual stresses and deformation [2,3].

SLM is a powder-based process, where a thin layer of powder is spread with a roller and heated with a laser beam following a scanning path. The molted powder cools down and consolidates building the layers of the part. After consolidation, the part lowers and a new layer of powder is applied. Due to the complexity of geometries and processes, SLM-produced parts often present undesired deformations or defects which limits the industrial development of such a technology [4]. The right set of process parameters can improve the part quality. For this reason, finding the best combination of laser power, speed, and path is a crucial aspect. Also, other parameters involving the characteristics of the powder, like the morphology and the size distribution, must be considered during the process parameter choice [5]. Manufacturers often rely on trial-and-error or empirical methods, such as design of experiment [6], to find a suitable set of parameters. Measurements of the final distortions are a very effective methods to establish the quality of the print [7]; however, experimental characterization can become time consuming and cost-expensive. Moreover, the results may be not extendable to different geometries [8]. Analytical methods can be a very useful tool to calculate temperatures and stresses and to correlate them with process and laser parameters to understand and improve the technology [9]. Although they are very suitable for simple features, they are not so efficient when working with complex geometries.

The thermal mechanical analysis for a laser bed fusion technology is fundamental to understanding the printing process and its parameters that determine the characteristics of the printed part. Several studies have been conducted to investigate single aspects involved in the SLM technologies such as melting pool, scan speed, size of the laser beam, build height effect, and inner layer time [10,11,12,13]. Those aspects are strictly correlated to the residual stresses that occur during the process and that produce the distortion of the final workpiece. To calculate such stresses, either an analytical or numerical approach can be used; the analytical approach can be useful to understand the physical parameters and can be easily adopted for simple cases. Among them, a reliable method considers the eigenfunctions approach [9] that can calculate the temperature and, therefore, the stresses that occur at different layers. Although the method gives a good prediction of the useful physical variable fields, it is more convenient to use a numerical approach like FEM (finite element method) when using complex geometries as is the case of this study.

Finite element simulation can be a powerful tool for process understanding and parameter identification. FEM allows the simulation of AM processes on different scales. Melt pool or mesoscale models can describe partial aspects of the process in detail [14,15], but FEM can also be used for macroscale models to understand the overall temperature progression and deformations during the process [16,17,18,19,20]. The heating process has been the subject of several studies [21,22,23,24,25] which allow us to understand more in detail the physical phenomena and to determine the relevance of the meso and microscale modeling. The microscale process has been investigated by Geng et al. in [21], where FE were combined with a microscopic phase field (PF) model that allowed the determination of the temperature distribution field and the effects of the microstructure evolution during a melting phase process for a wire arc additive manufacturing process (WAAM). The PF model can be combined with both FE and computational fluid dynamics (CFD), and it proved to be effective in evaluating the evolution of the chemical composition through concentration field equations and grain morphology, allowing for the determination of the quality of the microstructure by the temperature evolution. Moreover, it allows for the identification of the columnar dendritic spacing which can be useful to determine optimized process parameters such as laser speed and power. In [22], FE analysis was used by Cattenone et al. to determine the distortions and residual stresses at a meso and macroscale. This study investigated the process parameters and the modeling method through a workflow validated with experimental results. They determined the importance of constitutive material models as well as the meshing strategy and the time step in the local temperature distribution calculation. The FE analysis predicted the defects and the distortion of a manufactured object with fused deposition modeling (FDM) with a coefficient of variation of 12.2%. The study conducted in [25] with FE (using Abaqus AM module) simulated a laser direct energy deposition (LDED) AM process, and it allowed for the establishment of the importance of the surrounding powder bed and the relevance of the building plate thickness and its geometry constraint effects. In particular, the powder bed thickness became more relevant for small features, but, in such cases, the effect of the mesh needs to also be evaluated. The researchers also demonstrated that the time step of the thermal analysis can be incremental without compromising the final results, giving the possibility to save time on the calculation. Besides the modeling choices, the material characterization also resulted in playing a key role in the determination of accurate distortions as described in [22,25].

The simulation of large and complex geometries can result in excessive calculation times which limits its use for industrial applications. The direct modeling of laser scan lines requires a considerable number of nodes to be correctly represented and a high number of time increments. The findings presented in [26,27] reported run times of tens to hundreds of hours for modest-size models. In the last decade, the need for reliable and fast process parameter determination required the development of more efficient simulation methods. The common assumption is that the individual laser scan line modeling is no longer required, and approximations to simplify the analysis are used. Carraturo et al. used the finite cell method (FCM) to simulate the laser powder bed fusion (LPBF) process at part-scale by means of a layer-by-layer activation process. The simulation has been validated with publicly available experimental measurements of a single cantilever structure of Inconel 625 showing a maximum error of 4.72% and an almost perfect correlation with the experimental results. Hodge et al. [28] adopted a strategy called process agglomeration where the layers are not modeled at the scale of the physical powder but at a larger layer equal to 20 times the actual layer thickness. The model has been validated on stainless steel 316L, which are relatively small components [29] and required high-performance clusters. The surface deformations were measured via digital image correlation (DIC), while interior stresses were measured via neutron diffraction. The deformation results were described as good in terms of magnitude but with discrepancies in the distribution related to the model used. Another approach operates by activation of full layers or groups of layers at elevated temperatures. Often an analytic thermal load calculation determines the activation temperature of the layer and then a coupled thermo-mechanical calculation determines the mechanical response to the thermal loading. Zaeh and Branner [30] simulated the production of a T-shaped cantilever beam made with tool steel 1.2709 (X3NiCoMoTi18-9-5) and verified the deformations with experimental results obtained with a coordinate measuring machine (CMM). They showed that the method captured the trend of distortions, but the absence of a moving heat source model caused considerable overestimation of the peak distortion by 22.8%. Papadakis et al. [31] utilized a reduced thermal input method to predict the residual stress and distortion of an Inconel 718 cantilever. The maximum deviation between numerical and experimental results was 26%. Inherent strain models are another approach to AM process simulations. They rely on the assumption that the plastic strain developed during the process is uniform [32]. The two fundamental steps to solve inherent strain models are plastic strain calculation and plastic strain application [33], and usually the first step is achieved through experimental builds [34]. Another drawback of the inherent strain theory is the hypothesis of plastic strain homogeneity. With this method, any difference in plastic strain field caused by different geometries is neglected [34]. An advanced method for SLM simulation is multiscale modeling, where the result of the simulation of microscale physical phenomena is used as input for meso and macroscale simulations. Li et al. [35] described a multiscale model with different stages. In the microscale, a moving heat source was simulated to generate an integrated heat input for a mesoscale thermo-mechanical analysis of a larger volume for residual stress tensor calculation. The last stage consisted of mapping the residual stress tensor into a part-scale model. Despite the good agreement between simulation and experimental measurements, the model could not be extended to complex geometries. In another study, Li et al. [36] applied a multiscale model for the simulation of an AlSi10Mg cantilever beam production, reporting an error equal to 28% for the peak deformation.

Most of the models present in the literature consider only simple geometries. This allows a more straightforward results analysis but limits the understanding of the capabilities of FEM AM process simulation. Gauge et al. [37] considered a geometry with a complexity comparable to the one presented in this research work. They used a multi-scale model where the results of a small-scale analysis were used for the part-scale modeling and obtained good correlation with experimental results with a maximum of 13% for peak distortion and a minimum correlation of 90.5% for the chosen points. They considered a small thin-walled Inconel 625-compliant cylinder, a small Inconel 718 build with both very thin and very thick sections, and an industrial scale part formed from AlSi10Mg.

A workflow for AM system calibration and capability evaluation is presented in this research paper. ISO/ASTM 52902:2019 [38] covers the general description of benchmarking test piece geometries along with quantitative and qualitative measurements to be taken to assess the performance of AM systems. This work aims to expand the regulatory environment by introducing new geometries such as lattice structures, free forms, and cavities which are increasingly present in AM components. Moreover, the effectiveness of FE simulation for deformation prediction is shown alongside the precision of distortion measurement through iCT. A new calibration artifact (Figure 1) has been designed and manufactured with SLM technology and the deformations have been analyzed with iCT. A finite element simulation of the process has been performed, and the resulting deformations have been compared with the iCT results showing a good correlation. The geometry, the measurement method, and the simulation strategy are presented in the second section of this research work; the results are presented in the third section while the conclusions are outlined in the fourth and last sections.

## 2. Materials and Methods

### 2.1. Geometry and Production

The geometry considered in this research work is a platform that includes several features presented in [38] such as resolution slot (Figure 2), resolution pins and holes (Figure 3), circular artifact, cone, stairs, and linear artifact (Figure 4). The new approach consists of merging all the features and introducing some novel features that are increasingly present in AM parts but need to be evaluated with advanced technology such as iCT. Samples mentioned before could be analyzed with the standard measurements method such as coordinate-measuring machine (CMM) or centesimal caliber, due to their simple geometry (cubes, cylinders and circular holes, and flat surfaces). Freeform shapes (Figure 5), lattice structures (Figure 5), and cavities (Figure 6) have been produced to evaluate deformations and dimensional tolerances with a non-standard measurement method (iCT). The freeform shapes in this study are three 1.5 mm thickness metal sheets with different curvatures. There is also a horizontal freeform shape obtained below the flat surface of the plate (Figure 5b).

The material considered for the component production was an AlSi10Mg alloy powder 20–60 µm sourced from LPW Technology, Runcorn, UK, and printed with a layer thickness of 0.03 mm with one Yb (Ytterbium) fiber laser IR. The AlSi10Mg alloy has been chosen due to its popularity in the AM industry and the literature abundance of thermophysical properties data necessary for AM process simulation. The artifact has been manufactured with a Print Sharp 250 EP-M250 by Prima Additive, Torino, Italy, and with process parameters listed in Table 1. The process parameters have been determined by the manufacturer based on several tests and studies performed during their activity.

### 2.2. Thermal and Mechanical Analysis

A coupled thermoelastic analysis at the macro scale level is performed in the present activity. The governing equation for transient heat conduction is given as
(1)$$\nabla \left(-\kappa \nabla T\right)+\rho c\dot{T}=\rho h$$
(2)$$-\kappa \nabla T=q-h_{c}\left(T-T_{0}\right)-\sigma_{e}\varepsilon \left(T^{4}-T_{0}^{4}\right)$$
where T is the temperature, κ is the thermal conductivity of the material, ρ is the material density, h is the heat generation per unit of mass, q is the input heat flux, $h_{c}$ is the heat transfer coefficient under natural convection, $\sigma_{e}$ is the Stefan-Boltzmann constant, $T_{0}$ is the ambient temperature, and ε is the emissivity.

Deformations related to the final part have been calculated through a quasi-static mechanical analysis. The results of the thermal analysis constitute the thermal load for the mechanical one. The governing stress equilibrium equation is
(3)∇σ=0
where σ is the stress and follows Hook’s Law:(4)σ=Cε

C is the isotropic material stiffness tensor and ε is the total strain which includes elastic strain $\varepsilon_{e}$, plastic strain $\varepsilon_{p}$, and thermal strain $\varepsilon_{T}$. The thermal strain is computed as
(5)$$\varepsilon_{T}=\alpha \left(T\right)\left(T-T_{\mathrm{ref}}\right)$$
where $\alpha \left(T\right)$ is the temperature-dependent thermal expansion coefficient of the material and $T_{\mathrm{ref}}$ is the environment temperature.

### 2.3. Numerical Implementation

The FEM simulation is based on a macroscopic model of the SLM building process as implemented in the AMTOP^®^ V.2.0 software developed by ITACAe S.r.l, Asti, Italy, and SimTech Simulation et Technologie SARL, Paris, France. AMTOP^®^ V.2.0 is a platform of software tools developed to analyze and optimize additive manufacturing products and processes. The platform includes several algorithms to evaluate the extent of stresses and distortions through a “layer by layer” approach, consisting of loops of coupled thermal-structural analysis [39,40].

AMTOP^®^ V.2.0 calculates temperature, stress, and displacement history for different equivalent layers at the end of the bed powder manufacturing process. The equivalent layers are obtained as bundles of actual layers. Moreover, it computes the distortions that occur after the removal of the supports and the workpiece from the base plate. During the pre-processing phase, the manufacturing parameters are required as inputs: laser speed, laser power, laser path, layer thickness, hatch distance, the material used, the temperature of the plate, the temperature of the body, the temperature of the environment, the shape, and the dimensions of the supports. The latest is particularly important because they can determine how the distortion is created and can state the feasibility of the print. The AMTOP^®^ V.2.0 software can determine an optimized configuration of the supports to optimize a desired objective such as the height of the print or the volume of supports, but for the case considered, no support structures have been generated. The finite element analysis (FEA) is performed through an external solver (i.e., Calculix). The FEA consists of a series of thermal-mechanical simulations for each bundle of the printing process layer; the results of each bundle are the initial conditions of the next one. Such analyses allow for the determination of the history of the stress, displacement, and temperature field of the manufactured part. Once the simulated part is completely built, it is possible to determine the deformed geometry.

A scheme of the approach used is shown in Figure 7, and the correlation between the phases of the process and the corresponding modeling is shown in Table 2.

The accuracy of the simulation results strongly depends on the knowledge of the actual physical process parameters. Moreover, there are also numerical parameters that need to be set up to guarantee the best compromise between calculation time and accuracy. One of the important factors is the mesh size which is defined by the parameters beam diameter scale (DIASCALE) and layer thickness scale (LAYTHKSCALE) that determine the actual size of the mesh and the layer bundle height. One of the approach assumptions is to consider the size of the negligible laser dot compared to the characteristic dimensions of the component. In the preprocessing step, an algorithm prepares the FE model starting from the triangle tessellated boundary surface of the surface mesh of the part with a voxel meshing of the domain (Figure 8). The voxel sizes are multiples of DIASCALE and LAYTHKSCALE. The model is based upon the simplification that the laser energy is instantaneously absorbed into the system for each voxel layer. Therefore, the new layer elements are associated with a temperature higher than the melting temperature of the material considered.

The FE simulation of the selected geometry presented in this work requires further simplifications to give a result in terms of compatibility with industrial applications. The presence of the lattice structures implies a high number of elements to be correctly represented and, therefore, requires an excessive computational time. A strategy for simulation time reduction could be the use of equivalent homogeneous materials to simplify the geometry maintaining the thermo-mechanical properties of the lattice structure. A further approximation consisted of considering the lattice structures with bigger cell sizes as a void volume and the lattice structure with the smallest cell size as a solid volume with mechanical properties equal to the rest of the artifact (Figure 8). With this approximation, the deformations of the lattice structures cannot be obtained in the results of the simulation, and on the other hand, it consents to obtain the global deformation of the component in a few hours.

A first simulation (SIM1) has been conducted that neglects convection and radiation heat exchange and considers a perfect separation of the artifact from the base plate. Then a second simulation (SIM2) has been performed that considers convection heat transfer with $h_{c}=10\;W/m^{2}K$ [41] and includes the removal of 1.5 mm of material from the bottom surface represented by the removable elements in Figure 8. The process parameters considered for the simulation are the same used for the part production and are listed in Table 1. The thermophysical temperature-dependent properties of the AlSi10Mg alloy used for the simulation can be found in [42].

### 2.4. Industrial Computed Tomography

During an iCT scan using an X-ray system, multiple projections are taken systematically. The images are acquired from several different viewing angles obtained with the rotation of the sample. It is possible to obtain radiographic imaging due to different X-ray attenuation coefficients of materials, and the X-ray linear attenuation coefficients are represented as different iCT grey values. From these values, it is possible to obtain a virtual three-dimensional volume of a sample via reconstruction algorithms.

The obtained volume could be used for different purposes: One of the main applications of iCT volume is defect analysis, where all kinds of indications are analyzed looking for defects according to the requirements. iCT volume is also used in the field of metrology validation because it is the only non-destructive testing (NDT) technique that allows for having the full geometries of an internal feature. In this research, the NSI X5000 TEC Eurolab system by North Star Imaging, Rogers, MN, USA, was used for the metrological analysis of the designed artifact. The device is a Microfocus system with a Flat Panel detector, specifically designed to check components manufactured with light alloys or composite materials, for which a high resolution is required. iCT Volume could also be used for failure analysis, reverse engineering, and FEM simulation [43]. The scan was performed at 0.094 mm of resolution, 240 kV and 430 µA. After the surface calculation, the reference element and the analyzed surface/STL file were aligned through a best-fit registration.

## 3. Results and Discussion

The artifact manufactured with SLM technology is represented in Figure 9. An inspection of the component has been performed with a digital centesimal caliber, and the results are reported in Table 3; the dimension codes are reported in Figure 10.

The results of the iCT are reported in Figure 11, Figure 12, and Figure 13 where the volume obtained is compared to the nominal geometry (NG). The deviations with respect to the NG are generally smaller than 0.5 mm with higher values in the four corners and one of the vertical surfaces. Forty reference points have been considered to compare the different results, and the deviations are reported in Table 4 and Table 5. The component presents an upward concavity and the removal from the baseplate caused a loss of material at the base of the artifact as reported in Figure 12.

The simulation results have been compared to the iCT volume (Figure 14 and Figure 15) and to the NG (Figure 16). The first process simulation performed with AMTOP^®^ V.2.0 revealed differences in the deformations with respect to the iCT up to 1.20 mm as reported in Figure 14 and in Table 4. Moreover, the concavity of the bottom surface is opposite to the one observed in the manufactured part. For this reason, only 17 reference points have been considered, and the results of the first simulation have not been compared with the NG since the deformation pattern was clearly different. These differences can be attributed to the hypothesis of perfect separation from the baseplate and the neglect of convective heat exchange.

The second simulation considered convection heat exchange and the removal of 1.5 mm of material during the separation from the base plate to improve the fidelity with the manufactured part. In this case, the concavity was the same as observed with the iCT and the differences between the manufactured part and the predicted results are comparable and below 0.44 mm as reported in Figure 15 and in Table 5. Moreover, Figure 11 and Figure 16 present the comparison of the iCT and SIM2 with the initial geometry, respectively. The comparison revealed a common deformation pattern with similar deviations for the actual part and the digital twin as reported in Table 5. However, some differences are present, and the hypothesis considered for the lattice structures reduced considerably the calculation time, but the absence of the features determined some discrepancies in the results. In general, the numerical model and the voxel discretization introduce approximations that contribute to reducing the simulation accuracy; moreover, the comparison with the best-fitting alignment of the volumes can introduce small differences in the deviations.

The accuracy of the simulations can be quantified using two metrics, percent error of peak displacement, and by calculating the correlation over a field of representative points. The peak displacement evaluation would give an indication of how well the model predicts the most severe distortion. However, the base plate removal caused the loss of material, and for this reason, when comparing the iCT volume to the NG, the bottom of the artifact is the area indicated as the most distorted, but the discrepancies are not caused by the stress-induced deformations but by the absence of material. The same situation is present in the lattice structure area. For this reason, it is not possible to locate the actual peak distortion of the artifact.

On the other hand, correlation gives a more global indication of how accurate the model is. Correlation is calculated between two sets of data A and B with Equation (6)
(6)$$\mathrm{Correlation}\;\left(A,B\right)\%=\frac{\sum \left(a-\bar{a}\right)\left(b-\bar{b}\right)}{\sqrt{\sum {\left(a-\bar{a}\right)}^{2}\sum {\left(b-\bar{b}\right)}^{2}}}\times 100\%$$
where a and b are the members of the sets A and B, respectively, while $\bar{a}$ and $\bar{b}$ are the mean of A and B. The correlation between the two sets of data has been computed with Equation (6), revealing a value of 65%. This value does not reach the level of accuracy reported in other works. For example, Gouge et al. [37] reported a minimum correlation of 90.5% for a more sophisticated model. Considering the complexity of the studied geometry and the approximations introduced during the simulation phase, the correlation obtained is encouraging and can be further improved by increasing the number of reference points. The predicted and measured displacements taken at the 40 locations on the artifact surface are reported in Figure 17. The trend line represents the exact correspondence between predicted and measured displacements. The trend line plot indicates a good agreement between the simulation and the iCT volume.

## 4. Conclusions

The study reported in this research work presented an artifact geometry that included key features such as lattice structures, freeform structures, and cavities. The designed geometry has been manufactured with AlSi10Mg powder and a dimension inspection has been performed on most of the features with a centesimal caliber. The manufactured part has also been analyzed with an iCT. An FE model has been implemented to simulate the process. The model considered some simplifying hypotheses to reduce the computation time and, thus, make it suitable for industrial applications. These hypotheses included the lattice structure homogenization and the neglection of emissivity effects. A first simulation also considered no convection heat exchange and perfect separation of the part from the baseplate, while a second simulation considered convention and the removal of 1.5 mm of material from the bottom surface of the part to replicate the manufacturer’s cutting procedure. The comparison with the first simulation presented several differences in the points investigated and a different global deformation confirming the importance of convection heat exchange and a correct removal simulation. The second simulation revealed good accordance with a correlation equal to 65%. Thanks to the contribution of the software tool to the knowledge of the technology-based physical phenomenon, the implementation of the simulation methodology in the process engineering workflow can show several advantages to the manufacturer. These can be revealed in the reduction of time to market and costs, improvement of geometrical and structural properties, acceleration of the learning process, and the definition of a robust design methodology.

## Figures and Tables

**Figure 1 materials-16-04754-f001:**
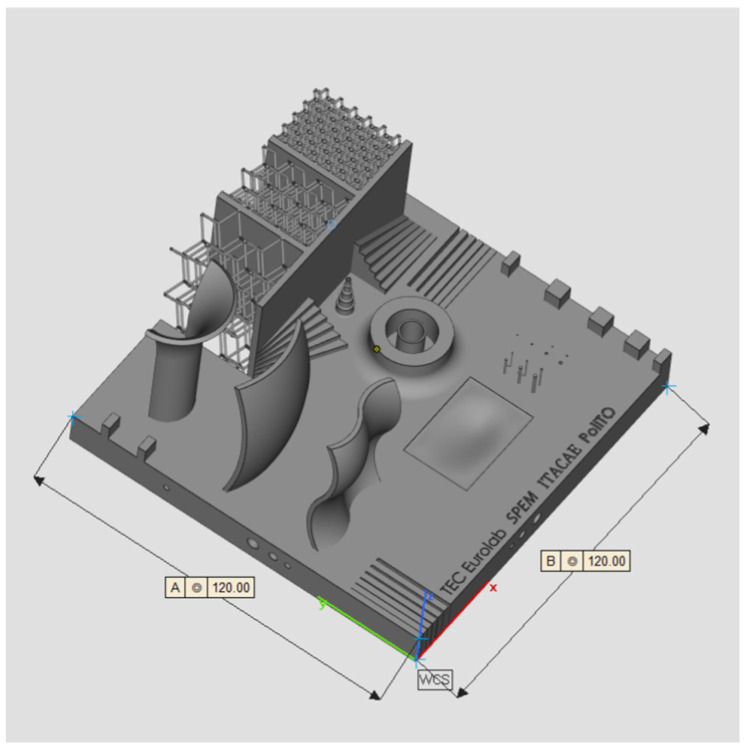
Calibrating artifact.

**Figure 2 materials-16-04754-f002:**
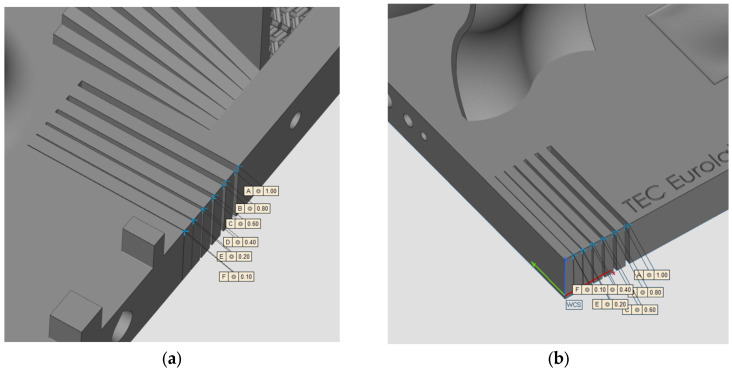
Resolution slot 1 (**a**); resolution slot 2 (**b**).

**Figure 3 materials-16-04754-f003:**
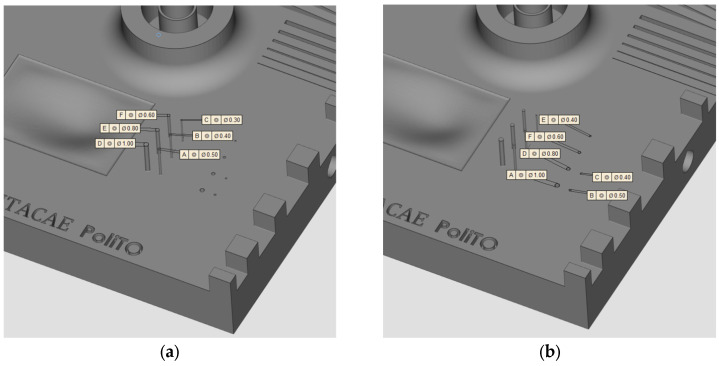
Pins with dimensions (**a**); holes with dimensions (**b**).

**Figure 4 materials-16-04754-f004:**
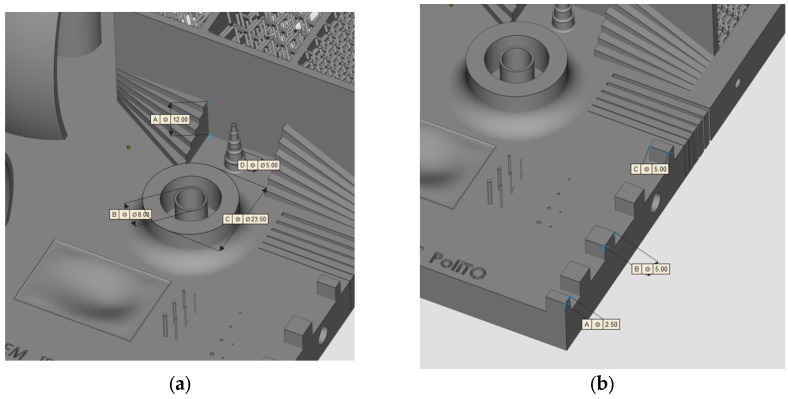
Circular artifact, stairs, and cone (**a**); linear artifact (**b**).

**Figure 5 materials-16-04754-f005:**
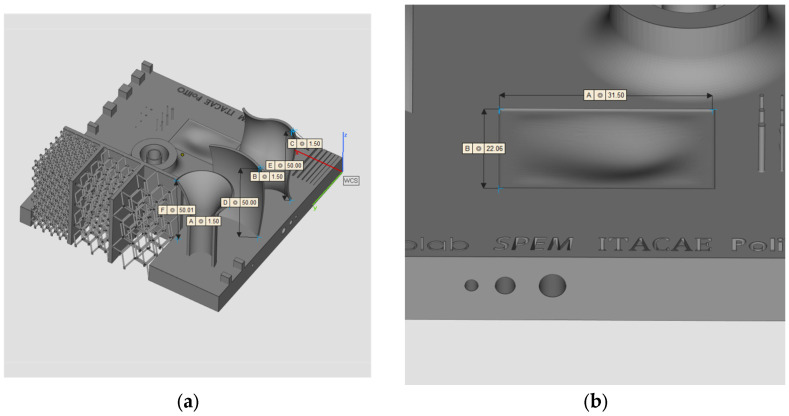
Vertical freeform (**a**); horizontal freeform (**b**).

**Figure 6 materials-16-04754-f006:**
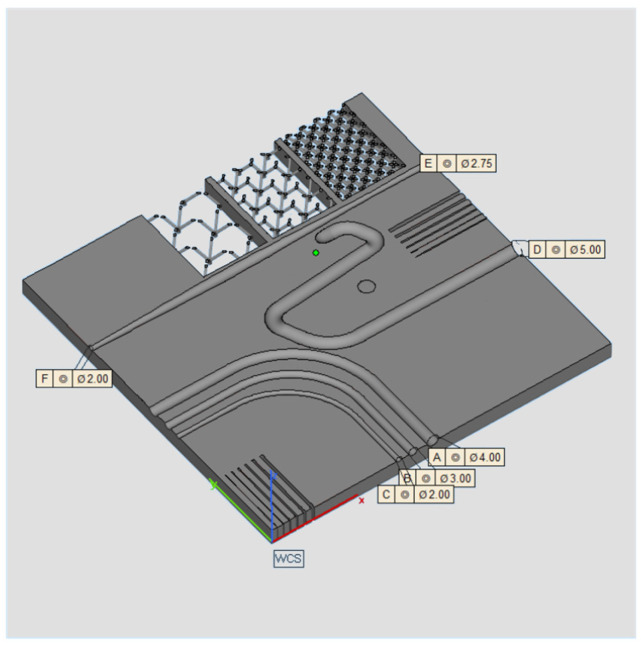
Cavities, section view.

**Figure 7 materials-16-04754-f007:**
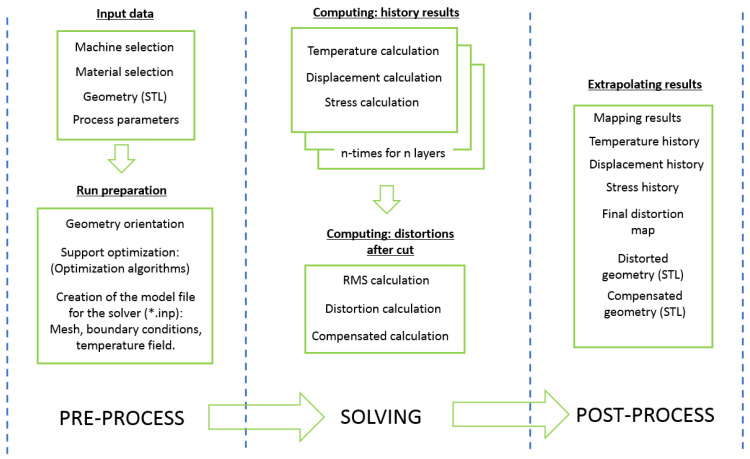
AMTOP^®^ V.2.0 Functioning Scheme.

**Figure 8 materials-16-04754-f008:**
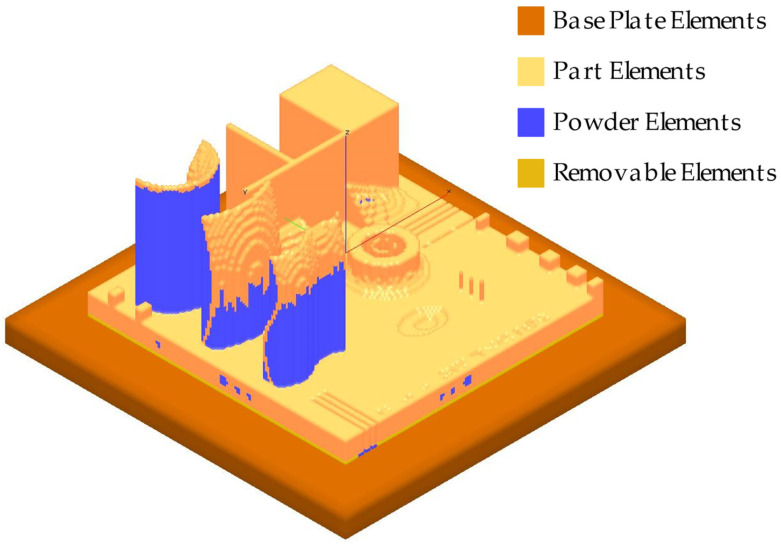
AMTOP^®^ V.2.0 Voxel Meshing.

**Figure 9 materials-16-04754-f009:**
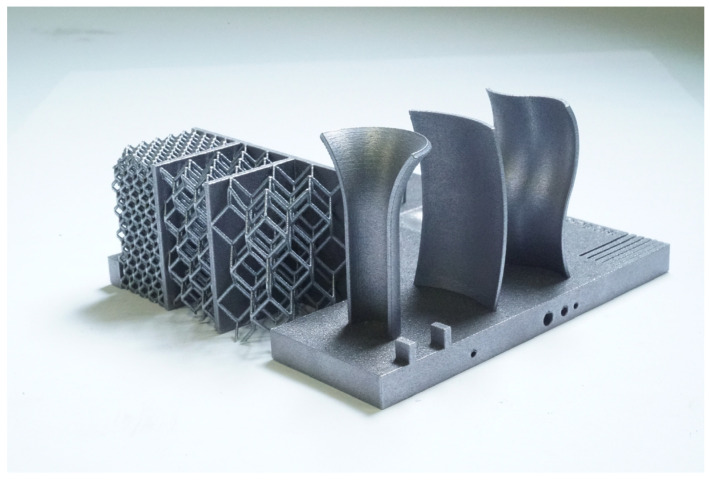
Artifact manufacturing result.

**Figure 10 materials-16-04754-f010:**
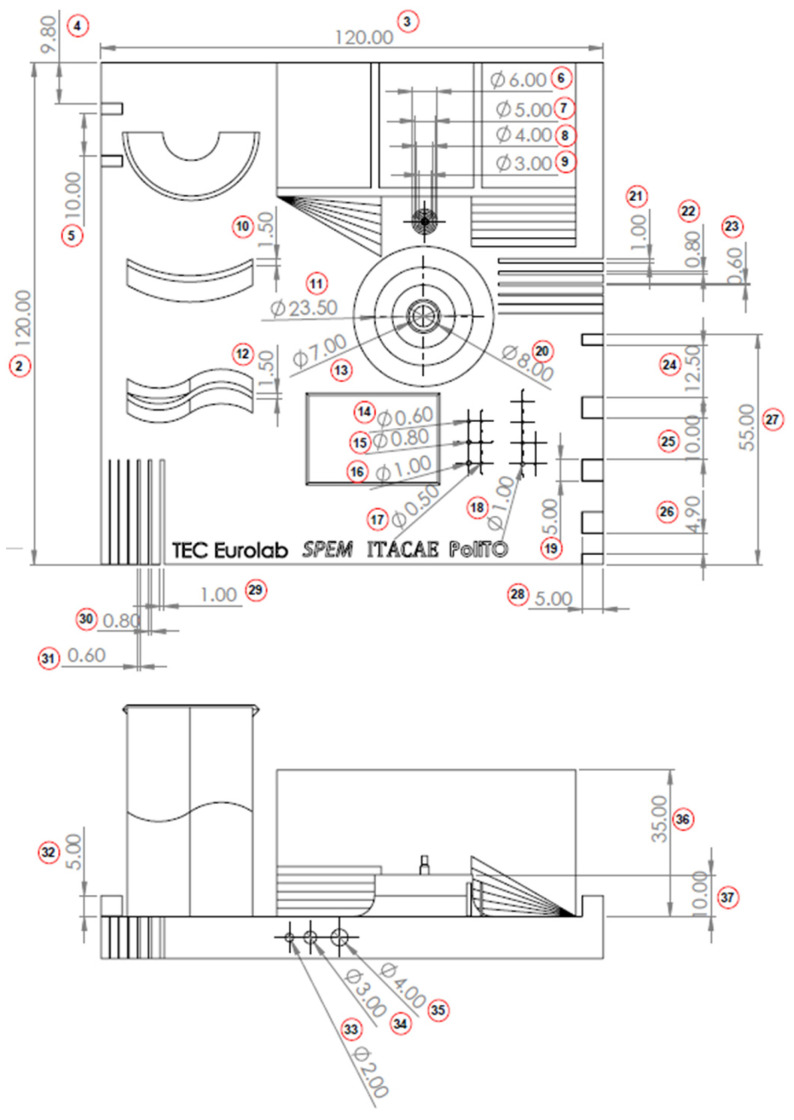
Artifact-inspected dimensions.

**Figure 11 materials-16-04754-f011:**
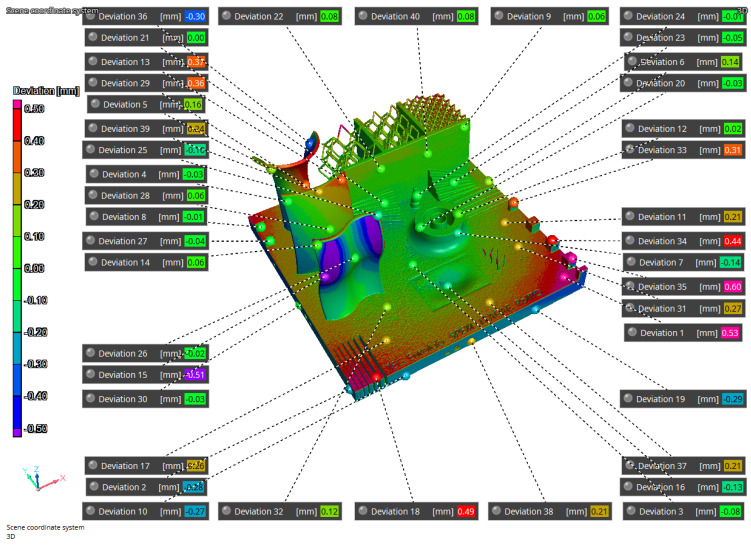
NG-iCT comparison, top view.

**Figure 12 materials-16-04754-f012:**
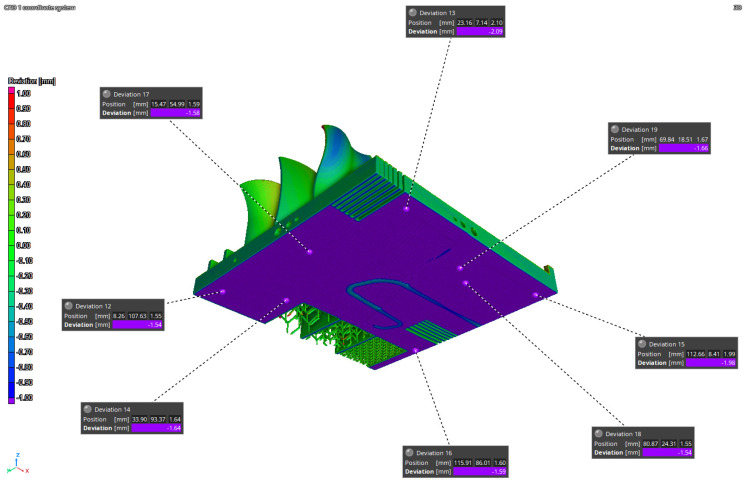
NG-iCT comparison, bottom view.

**Figure 13 materials-16-04754-f013:**
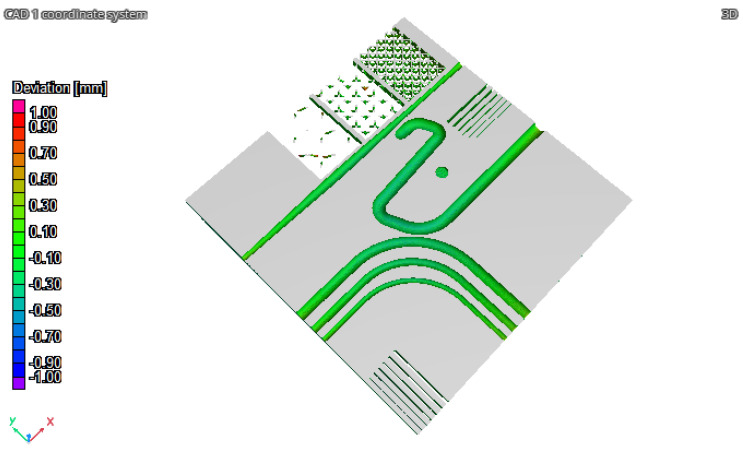
NG-iCT comparison, internal view.

**Figure 14 materials-16-04754-f014:**
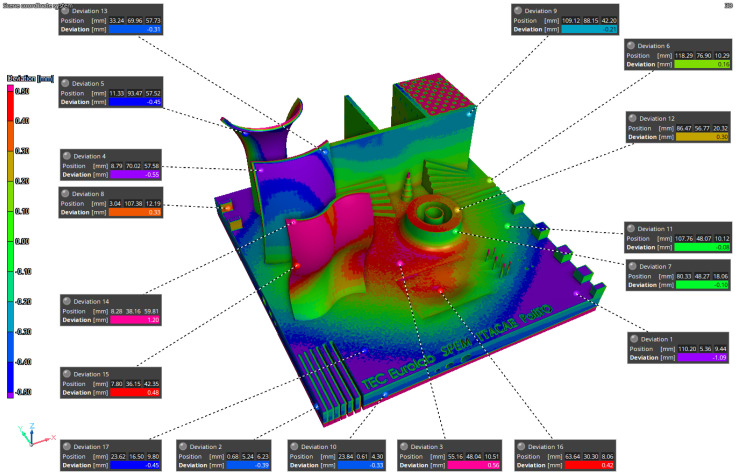
SIM1-iCT comparison.

**Figure 15 materials-16-04754-f015:**
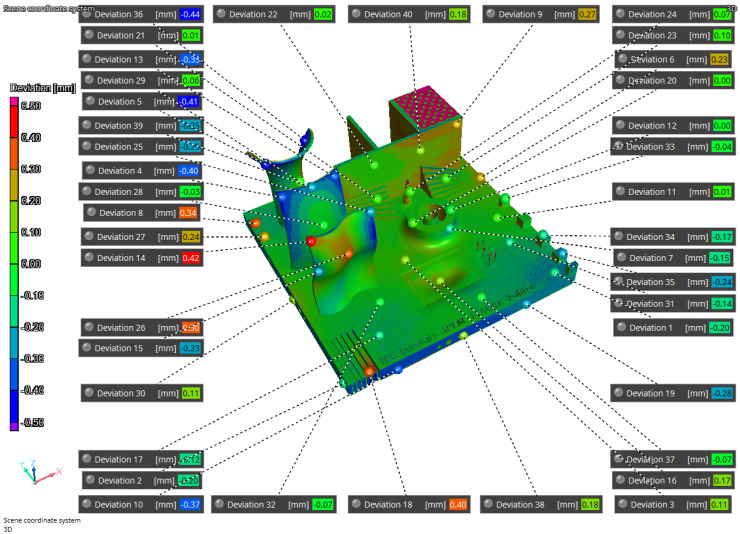
SIM2-iCT comparison.

**Figure 16 materials-16-04754-f016:**
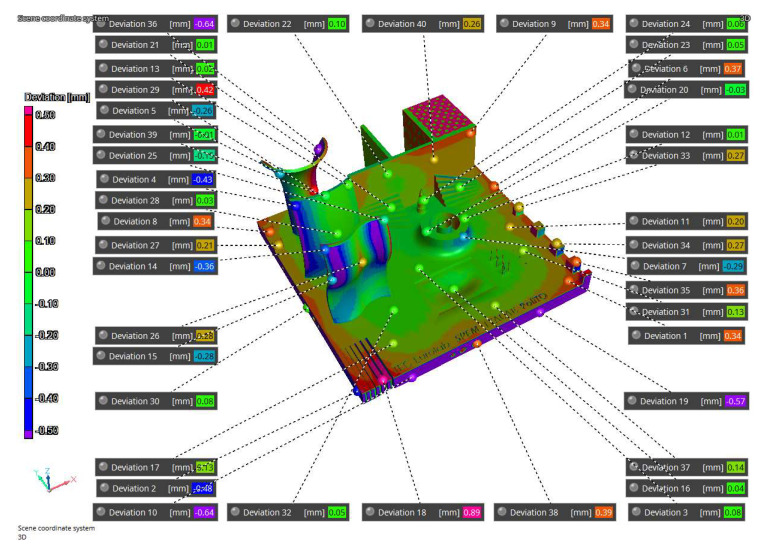
SIM2-NG Comparison.

**Figure 17 materials-16-04754-f017:**
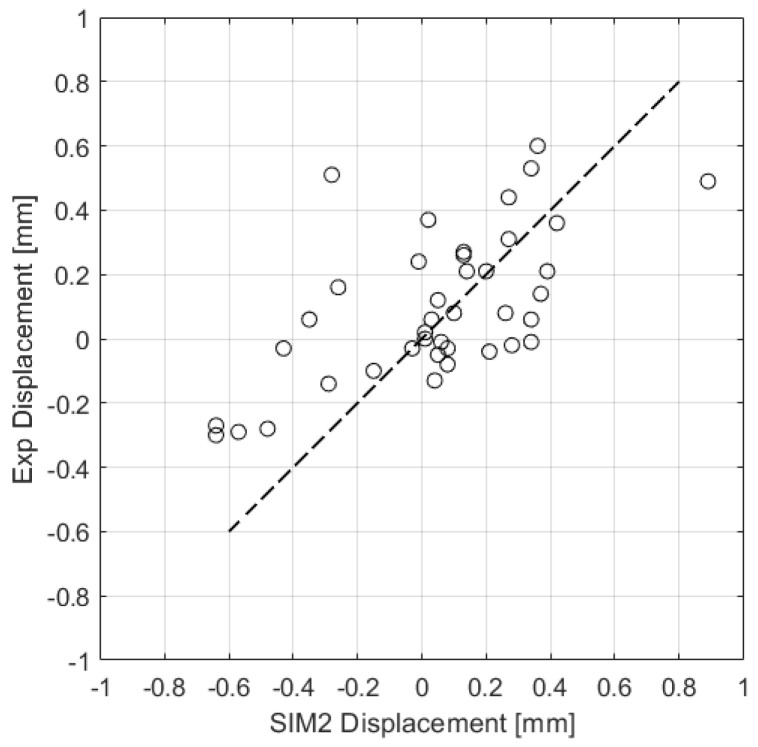
Comparison of 40 individual simulation-measurement points.

**Table 1 materials-16-04754-t001:** Process parameters.

Property	Value
Laser Power	370 W
Platform Temperature	120 °C
Scan Speed	1300 mm/s
Scan Strategy	Stripe of 10 mm
N° of contour	2
Spot (Laser)	0.11 mm
Gas type	$N_{2}$

**Table 2 materials-16-04754-t002:** Modeling of the physical process.

Process Phase	Simulation Phase
Part orientation and placement	AMTOP^®^ V.2.0 can suggest the best orientation strategy. Process parameters need to be assigned and a mesh sensitivity study performed.
Layer 1: fusion of the powder for the first layer bundle	The model calculates the temperature of the first bundle of layers. The model requires the temperature field as the initial condition for the FEM solver and the geometry of the first layer bundle
Recoating: deposition of the powder for the next layer	Recoating time and cooldown time are considered in this phase
Layer 1+n: fusion of the powder for the first layer	At each layer, the calculation provides stress, displacement, and temperature field that depends on the results of the previous layer which are the initial condition of the current one.
Cutting from the plate and the supports	The cutting removes elements (supports + plate) that do not belong to the printed part. In this phase, AMTOP^®^ V.2.0 calculates the final distortion.

**Table 3 materials-16-04754-t003:** Results of dimensions inspection.

Code	Dimension to Check [mm]	Inspection Results [mm]	%Difference
1	\	\	\
2	120.00	119.59	0.34
3	120.00	119.66	0.28
4	9.80	9.70	1.02
5	10.00	9.86	1.4
6	Ø6.00	ND-NA	/
7	Ø5.00	Ø5.01	0.2
8	Ø4.00	Ø3.95	1.25
9	Ø3.00	Ø2.98	0.67
10	1.50	1.55	3.33
11	Ø23.50	23.68	0.77
12	1.50	1.54	2.67
13	Ø7.00	6.84	2.29
14	Ø0.60	Ø0.68	13.3
15	Ø0.80	Ø0.79	1.25
16	Ø1.00	Ø0.95	5
17	Ø0.50	Ø0.58	16
18	Ø1.00	Ø0.88	12
19	5.00	5.01	0.2
20	Ø8.00	Ø7.98	0.25
21	1.00	0.99	1
22	0.80	0.77	3.75
23	0.60	0.58	3.33
24	12.50	12.99	3.92
25	10.00	9.89	1.1
26	4.90	4.86	0.82
27	55.00	54.93	0.127
28	5.00	4.99	0.2
29	1.00	0.93	7
30	0.80	0.77	3.75
31	0.60	0.56	6.67
32	5.00	5.12	2.4
33	Ø2.00	1.78 × 1.60 (elliptic)	20
34	Ø3.00	2.84 × 2.58 (elliptic)	14
35	Ø4.00	3.80 × 3.31 (elliptic)	5
36	35.00	35.08	0.23
37	10.00	10.09	0.9

**Table 4 materials-16-04754-t004:** Results of iCT volume comparison with respect to SIM1.

Point	iCT-SIM1 [mm]
1	−1.09
2	−0.39
3	0.56
4	−0.55
5	−0.45
6	0.16
7	−0.10
8	0.33
9	−0.21
10	−0.33
11	−0.08
12	0.30
13	−0.31
14	1.20
15	0.48
16	0.42
17	−0.45

**Table 5 materials-16-04754-t005:** Results of iCT volume comparison with respect to the initial geometry and with respect to SIM2 results.

Point	NG-iCT [mm]	NG-SIM2 [mm]	iCT-SIM2 [mm]
1	0.53	0.34	−0.20
2	−0.28	−0.48	−0.20
3	−0.08	0.08	0.11
4	−0.03	−0.43	−0.40
5	0.16	−0.26	−0.41
6	0.14	0.37	0.23
7	−0.14	−0.29	−0.15
8	−0.01	0.34	0.34
9	0.06	0.34	0.27
10	−0.27	−0.64	−0.37
11	0.21	0.20	0.00
12	0.02	0.01	0.00
13	0.37	0.02	−0.35
14	0.06	−0.35	0.42
15	0.51	−0.28	−0.23
16	−0.13	0.04	0.17
17	0.26	0.13	−0.12
18	0.49	0.89	0.40
19	−0.29	−0.57	−0.28
20	−0.03	−0.03	0.00
21	0.00	0.01	0.01
22	0.08	0.10	0.02
23	−0.05	0.05	0.10
24	−0.01	0.06	0.07
25	−0.10	−0.15	−0.25
26	−0.02	0.28	0.30
27	−0.04	0.21	0.24
28	0.06	0.03	−0.03
29	0.36	0.42	0.06
30	−0.03	0.08	0.11
31	0.27	0.13	−0.14
32	0.12	0.05	−0.07
33	0.31	0.27	−0.04
34	0.44	0.27	−0.17
35	0.60	0.36	−0.24
36	−0.30	−0.64	−0.44
37	0.21	0.14	−0.07
38	0.21	0.39	0.18
39	0.24	−0.01	−0.25
40	0.08	0.26	0.18

## Data Availability

Data available on request.

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
