# Peer review of "FEM Simulation of AlSi10Mg Artifact for Additive Manufacturing Process Calibration with Industrial-Computed Tomography Validation"

_materials, 2023, doi:10.3390/ma16134754_

Round 1
Reviewer 1 Report
The paper is largely complete and well-prepared. However, it would be appropriate to review the following points.
How were the geometries and dimensions specified in Section 2.1 determined?
Why AlSi10Mg material was chosen
More information should be given about the optimization work carried out. Which method was chosen, what are the parameters etc.
Which solvent was used in FEA? A little more detailed information about FEA should be added.
Author Response
How were the geometries and dimensions specified in Section 2.1 determined?
Part of the features are described in ISO/ASTM 62902:2019. These features include resolution slots, pins, holes, circular artifact, stairs, cone, and linear artifact. The innovative features are the cavities, the freeform, and the lattice structure which have been introduced to expand the regulation with geometries that are increasingly present in AM components. The dimensions of the features were limited by the boundaries of the Computed Tomography Scan and have been chosen in order to be comparable with the other feature described in ISO/ASTM.
Why AlSi10Mg material was chosen
AlSi10Mg is one of the most common alloys used in AM production, moreover, in literature there is an abundance of thermophysical properties related to this material which consent to having reliable data for AM process simulation. These reasons are now specified in the text.
More information should be given about the optimization work carried out. Which method was chosen, what are the parameters etc.
Thank you for pointing out the relevance of optimization in AM processing. This work aims to demonstrate the capabilities of a FE AM process simulation in predicting final distortions. For the manufacturing of the calibrating artifact, the process parameters used were not optimized with FEA but were determined by the manufacturer on the basis of several tests and studies performed during their activity. After validation of this procedure, a further step for the research could be optimizing the process parameters through FE simulation or the definition of a counter-deformed geometry.
Which solvent was used in FEA? A little more detailed information about FEA should be added.
The solver used is Calculix as specified in the text. Calculix is a free solver chosen in order to be not bounded by licensed software. However, the input file can also be read with other solvers like ABAQUS. The reference for the thermophysical properties used for FEA is now reported in the text.
Reviewer 2 Report
The article is interesting but needs to be rewritten.
The introduction must be corrected. Only general informations were provided. Authors should describe 10-20 papers focusing on similar topics. For each of the papers the information about what was tested, testing method, used materials, results, and advantages/disadvantages of the presented approach should be provided. The numerical values should be presented for each of the paper mentioned
The conclusion section should be precise with the numerical values provided. The results should be compared to the studied literature.
The IMRAD structure should be used.
The article is interesting but needs to be rewritten.
The introduction must be corrected. Only general informations were provided. Authors should describe 10-20 papers focusing on similar topics. For each of the papers the information about what was tested, testing method, used materials, results, and advantages/disadvantages of the presented approach should be provided. The numerical values should be presented for each of the paper mentioned
The conclusion section should be precise with the numerical values provided. The results should be compared to the studied literature.
The IMRAD structure should be used.
Author Response
The article is interesting but needs to be rewritten.
The introduction must be corrected. Only general informations were provided. Authors should describe 10-20 papers focusing on similar topics. For each of the papers the information about what was tested, testing method, used materials, results, and advantages/disadvantages of the presented approach should be provided. The numerical values should be presented for each of the paper mentioned
The conclusion section should be precise with the numerical values provided. The results should be compared to the studied literature.
The authors thank the reviewer for the encouraging words and for helping us improve the quality of the manuscript. Several previous works have been analyzed and now the discussion concerning their findings has been extended. Some references are reported below with a brief description of the results obtained. Due to the novelty of the research field, the number of works similar to the submitted manuscript is limited. Moreover, most of the previous work focused on the temperature distribution prediction of small components or single layers. However, some relevant examples of part-scale simulation with deformation measurement exist. Some of them involve very simple geometries and just a few of them analyze components with a complexity comparable to the presented calibrating artifact.
Papadakis, A. Loizou, J. Risse, e J. Schrage, «Numerical Computation of Component Shape Distortion Manufactured by Selective Laser Melting», Procedia CIRP, vol. 18, pp. 90–95, 2014, doi: 10.1016/j.procir.2014.06.113. In this work, the authors utilized the reduced thermal input method to predict the residual stress and distortion of an Inconel 718 cantilever. The maximum deviation between numerical and experimental results was 26%
Hodge, N.E.; Ferencz, R.M.; Vignes, R.M. (2016). Experimental Comparison of Residual Stresses for a Thermomechanical Model for the Simulation of Selective Laser Melting. Additive Manufacturing, (), S221486041630094X–. doi:10.1016/j.addma.2016.05.011. Hodge et al. used a part-scale model described in (“A.S. Wu, D.W. Brown, M. Kumar, G.F. Gallegos, and W.E. King. An Experimental Investigation into Additive Manufacturing-Induced Residual Stresses in 316L Stainless Steel. Metallurgical and Materials Transactions A, 45(13):6260–6270, September 2014.”) for the production simulation of 316L stainless steel components. They measured surface deformations associated with the sectioning of the part, measured via digital image correlation (DIC), and interior stresses, measured via neutron diffraction. The correlation between experimental and numerical results was good in terms of magnitude but less in terms of distribution with discrepancies related to the model used. No detailed indications of the accuracy of the results is given.
Michael F. Zaeh; Gregor Branner (2010). Investigations on residual stresses and deformations in selective laser melting. , 4(1), 35–45. doi:10.1007/s11740-009-0192-y. Zaeh et. al simulated a T-shaped cantilever beam with different alloys and verified the deformations with the coordinate measuring machine and the stresses with neutron diffraction. The results revealed some discrepancies, with a maximum deformation equal to -0.386 mm against the predicted -0.50mm (error 22.8%)
Carraturo, Massimo; Jomo, John; Kollmannsberger, Stefan; Reali, Alessandro; Auricchio, Ferdinando; Rank, Ernst (2020). Modeling and experimental validation of an immersed thermo-mechanical part-scale analysis for laser powder bed fusion processes. Additive Manufacturing, 36(), 101498–. doi:10.1016/j.addma.2020.101498. The authors used the finite cell method (FCM) to simulate the LPBF process at part-scale by means of a layer-by-layer activation process. The simulation has been validated with publicly available experimental measurements of a single-cantilever structure of Inconel 625 showing a maximum error of 4.72% and an almost perfect correlation with the experimental results.
Bayat, Mohamad; Klingaa, Christopher G.; Mohanty, Sankhya; De Baere, David; Thorborg, Jesper; Tiedje, Niels S.; Hattel, Jesper H. (2020). Part-scale thermo-mechanical modeling of distortions in Laser Powder Bed Fusion â Analysis of the sequential flash heating method with experimental validation. Additive Manufacturing, (), 101508–. doi:10.1016/j.addma.2020.101508 Used the Sequential Flash Heating (SFH) method for a part-scale simulation of LPBF process of different Ti6Al4V specimens. The model overestimated the minimum deflection magnitude by 46.2%, the error was considerably reduced to 1.19% with mesh refinement, but the computational time was 65 hours.
Li, C.; Fu, C.H.; Guo, Y.B.; Fang, F.Z. (2016). A multiscale modeling approach for fast prediction of part distortion in selective laser melting. Journal of Materials Processing Technology, 229(), 703–712. doi:10.1016/j.jmatprotec.2015.10.022. Li et al.described a multiscale model with different stages. Despite the good agreement between the simulation of a thin rectangular iron object production and experimental measurements, the model could not be extended to complex geometries
Li, J. F. Liu, X. Y. Fang, e Y. B. Guo, «Efficient predictive model of part distortion and residual stress in selective laser melting», Addit. Manuf., vol. 17, pp. 157–168, ott. 2017, doi: 10.1016/j.addma.2017.08.014. applied a multiscale model for the simulation of AlSi10Mg cantilever beam production reporting an error equal to 28% for the peak deformation.
Most of the models present in the literature consider only simple geometries. This consent a more straightforward results analysis but limits the understanding of the capabilities of FEM AM process simulation. Moreover, the geometry presented in the manuscript undergoes a best-fitting orientation when the predicted or manufactured deformed artifacts are compared to the nominal geometries. This orientation strategy can vary and can produce different deviations.
The work of Gauge et. al. M. Gouge, E. Denlinger, J. Irwin, C. Li, e P. Michaleris, «Experimental validation of thermo-mechanical part-scale modeling for laser powder bed fusion processes», Addit. Manuf., vol. 29, p. 100771, ott. 2019, doi: 10.1016/j.addma.2019.06.022. considered a geometry with a complexity comparable to the one presented in this research work. They used a multi-scale model where the results of a small-scale analysis are used for the part-scale modeling obtaining good correlation with experimental results with a maximum of 13% for peak distortion and a minimum correlation of 90.5% for the chosen points. They considered a small thin-walled Inconel 625 Compliant Cylinder, a small Inconel 718 build with both very thin and very thick sections, and an industrial scale part formed from AlSi10Mg.
In the presented research, the maximum deviations fall into the area where the material was cut during the base plate removal. In this case, the deviations are not caused by a stress-induced deformation but by the absence of material and therefore can’t be considered for the determination of the deformation error. On the other hand, the number of points has been increased to compute the correlation between numerical and experimental points according to the procedure indicated by Gauge et al. The correlation presented a value of 65% for the chosen points which is encouraging considering the simplification introduced with the numerical model and the mesh size.
The IMRAD structure should be used.
The Results section has been changed into a Results and Discussion section while the Conclusion section has been shortened.
Reviewer 3 Report
The article highlights the features of modelling the FE of the AlSi10Mg artifact to evaluate the additive manufacturing process with the verification of industrial computed tomography. The authors simulated the FE process taking into account material removal during base plate separation, and compared the calculated distortions with iCT results. A good match between the final product and its digital twin is shown. The authors did not pay the necessary attention to the material (powder) from which artifact was made.
The article is interesting, but a number of shortcomings need to be corrected:
1. Along with the statement that “The right set of process parameters can improve the part quality…”, the authors should note that the size of the powder in Selective Laser melting also affects (For example, https://doi.org/10.1007/s11106-019-00033-8).
2. Authors should indicate what is the difference between Fig. 1a and Fig. 1b (no need to give two Figs. that differ in the presence or absence of dimensions), if there is no difference, then you need to remove Fig. 1a.
3. Authors should indicate what is the difference between Fig. 6a and Fig. 6b (no need to give two Figs. that differ in the presence or absence of dimensions), if there is no difference, then you need to remove Fig. 6a.
4. The authors should indicate from what considerations they chose the process parameters indicated in Table 1.
5. Please state manufacturer, city and country from where equipment has been sourced. This has to be done for each equipment, software, material and chemical in the paper.
Author Response
The article highlights the features of modelling the FE of the AlSi10Mg artifact to evaluate the additive manufacturing process with the verification of industrial computed tomography. The authors simulated the FE process taking into account material removal during base plate separation, and compared the calculated distortions with iCT results. A good match between the final product and its digital twin is shown. The authors did not pay the necessary attention to the material (powder) from which artifact was made.
The article is interesting, but a number of shortcomings need to be corrected:
- Along with the statement that “The right set of process parameters can improve the part quality…”, the authors should note that the size of the powder in Selective Laser melting also affects (For example, https://doi.org/10.1007/s11106-019-00033-8).
Thank you for helping us improve the quality of our manuscript. The introduction has been modified including these observations.
- Authors should indicate what is the difference between Fig. 1a and Fig. 1b (no need to give two Figs. that differ in the presence or absence of dimensions), if there is no difference, then you need to remove Fig. 1a.
- Authors should indicate what is the difference between Fig. 6a and Fig. 6b (no need to give two Figs. that differ in the presence or absence of dimensions), if there is no difference, then you need to remove Fig. 6a.
Thank you for pointing out these issues, the figures have been replaced.
- The authors should indicate from what considerations they chose the process parameters indicated in Table 1.
The process parameters have been determined by the manufacturer on the basis of several tests and studies performed during their activity. One of the possible applications of AM process simulation is the optimization of process parameters but in this case, the manufacturer indications have been followed.
- Please state manufacturer, city and country from where equipment has been sourced. This has to be done for each equipment, software, material and chemical in the paper.
The information required is now reported in the manuscript.
Reviewer 4 Report
In this manuscript, a geometry for process parameters calibration is presented. The part has been manufactured and then analyzed with industrial computed tomography. A finite element simulation of the process has been performed considering material removal during base plate separation and the resulting deformation has been compared with the results of the iCT. The work is meaningful and interesting for additive manufacturing community.
Overall, the paper is more like a technical report than an academic paper. The comparison between the FE simulation and industrial computed tomography were presented. However, an analysis of the mechanism behind the results is lacking.
The content for Section 3 “Result”is not enough for an academic paper. The depth of the work is not enough and needs to be further improved.
For Conclusion Section, it is too long. In this Section, the important results of the work should be refined and summarized.
In Table 3, how was the “iCT - 1st Sim.”derived? The results for “1st Sim.” were not given.
Line 133, “1,5 mm”should be “1.5 mm”.
Line 144, “0,03 mm”should be “0.03 mm”.
In Table 1, “0,11mm” should be “0.11mm”.
Line 234, “1,5 mm”should be “1.5 mm”.
Line 292, 1,5 mm”should be “1.5 mm”
Line 331, 1,5 mm”should be “1.5 mm”
Line 43, “After consolidation the part lowers and a new 43 layer of powder is applied”. A comma is missing.
Line 46, “The right set of process parameters can improve the part quality, for this reason finding the best combination of laser power, speed, and path is a crucial aspect.”should be changed as “The right set of process parameters can improve the part quality. For this reason, finding the best combination of laser power, speed, and path, is a crucial aspect.”
Author Response
In this manuscript, a geometry for process parameters calibration is presented. The part has been manufactured and then analyzed with industrial computed tomography. A finite element simulation of the process has been performed considering material removal during base plate separation and the resulting deformation has been compared with the results of the iCT. The work is meaningful and interesting for additive manufacturing community.
The author thanks the reviewer for recognizing the importance of the presented research work for AM community.
Overall, the paper is more like a technical report than an academic paper. The comparison between the FE simulation and industrial computed tomography were presented. However, an analysis of the mechanism behind the results is lacking.
The content for Section 3 “Result”is not enough for an academic paper. The depth of the work is not enough and needs to be further improved.
For Conclusion Section, it is too long. In this Section, the important results of the work should be refined and summarized.
To answer the previous three points, the typesetting of the manuscript has been modified extending the Result section which is now “results and discussion” and reducing the size of the conclusion section. Moreover, the number of reference points has been increased and the correlation between numerical and experimental results is now reported and compared to the results obtained in similar publications.
In Table 3, how was the “iCT - 1st Sim.”derived? The results for “1st Sim.” were not given.
The column “iCT - 1st Sim” report the same results indicated in Figure 16 (now 15) which presents the comparison between the deformation predicted during the first simulation and the iCT volume. Due to the big differences highlighted by this comparison, the results of the first simulation have not been compared with the Nominal Geometry. These considerations are now reported in the text.
Line 133, “1,5 mm”should be “1.5 mm”.
Line 144, “0,03 mm”should be “0.03 mm”.
In Table 1, “0,11mm” should be “0.11mm”.
Line 234, “1,5 mm”should be “1.5 mm”.
Line 292, 1,5 mm”should be “1.5 mm”
Line 331, 1,5 mm”should be “1.5 mm”
Line 43, “After consolidation the part lowers and a new 43 layer of powder is applied”. A comma is missing.
Line 46, “The right set of process parameters can improve the part quality, for this reason finding the best combination of laser power, speed, and path is a crucial aspect.”should be changed as “The right set of process parameters can improve the part quality. For this reason, finding the best combination of laser power, speed, and path, is a crucial aspect.”
The authors thank the reviewer for the meticulous review of the manuscript. The mistakes indicated in all the previous points have been corrected.
Reviewer 5 Report
Figures 7 and 10 should be omitted from the main body. You may insert them into an additional file or use with the graphical abstract. Add appropriate information with the section.
Section 2.1 - changes should be made to bring more clarity on the numerical simulations performed by bringing forth specific data deployed
Data in Table 2 should be revised for conformity.
Author Response
Figures 7 and 10 should be omitted from the main body. You may insert them into an additional file or use with the graphical abstract. Add appropriate information with the section.
The figures have been omitted from the main body and further information on the devices manufacturers are now present in the text.
Section 2.1 - changes should be made to bring more clarity on the numerical simulations performed by bringing forth specific data deployed
The data used during the simulation have been completed with the reference used for the AlSi10Mg thermophysical properties, which can be found in “X. Zhang, J. Kang, Y. Rong, P. Wu, e T. Feng, «Effect of Scanning Routes on the Stress and Deformation of Overhang Structures Fabricated by SLM», Materials, vol. 12, fasc. 1, p. 47, dic. 2018, doi: 10.3390/ma12010047.” Now the manuscript should report all the processes and material data used for the simulation. The authors hope that the data provided match the reviewer's expectations.
Data in Table 2 should be revised for conformity.
Table 2 has been revised for conformity, the authors wish to thank the reviewer for helping us improve the quality of the manuscript
Round 2
Reviewer 1 Report
The authors have made relevant corrections.
Reviewer 2 Report
I recommend to publish the article
I recommend to publish the article
Reviewer 3 Report
The authors took into account all comments of the reviewer and made appropriate corrections to the manuscript.
Reviewer 4 Report
The manuscript was improved a lot. There are still some English problems, especially for the newly added paragraphs. It is advised to check the whole manuscript carefully when proofreading.
Line 51, “Measurements of the final distortions are a very effective methods to establish the quality of the print” should be corrected as “Measurement of the final distortion is a very effective method to establish the quality of the print”
Line 78, “FE were combined with a microscopic phase field (PF) model” should be corrected as “FE were combined with a microscopic phase field (PF) model”.
Reviewer 5 Report
Dear esteem researchers,
I would like to congratulate on your endeavor.